# The Impact of Single-Cell Genomics on Adipose Tissue Research

**DOI:** 10.3390/ijms21134773

**Published:** 2020-07-05

**Authors:** Alana Deutsch, Daorong Feng, Jeffrey E. Pessin, Kosaku Shinoda

**Affiliations:** 1Departments of Medicine, Albert Einstein College of Medicine, Bronx, New York, NY 10461, USA; alana.deutsch@einsteinmed.org (A.D.); daorong.feng@einsteinmed.org (D.F.); jeffrey.pessin@einsteinmed.org (J.E.P.); 2Departments of Molecular Pharmacology, Albert Einstein College of Medicine, Bronx, New York, NY 10461, USA; 3Fleischer Institute of Diabetes and Metabolism, Albert Einstein College of Medicine, Bronx, New York, NY 10461, USA

**Keywords:** obesity, adipocyte, adipose, genomics, transcriptome, molecular metabolism

## Abstract

Adipose tissue is an important regulator of whole-body metabolism and energy homeostasis. The unprecedented growth of obesity and metabolic disease worldwide has required paralleled advancements in research on this dynamic endocrine organ system. Single-cell RNA sequencing (scRNA-seq), a highly meticulous methodology used to dissect tissue heterogeneity through the transcriptional characterization of individual cells, is responsible for facilitating critical advancements in this area. The unique investigative capabilities achieved by the combination of nanotechnology, molecular biology, and informatics are expanding our understanding of adipose tissue’s composition and compartmentalized functional specialization, which underlie physiologic and pathogenic states, including adaptive thermogenesis, adipose tissue aging, and obesity. In this review, we will summarize the use of scRNA-seq and single-nuclei RNA-seq (snRNA-seq) in adipocyte biology and their applications to obesity and diabetes research in the hopes of increasing awareness of the capabilities of this technology and acting as a catalyst for its expanded use in further investigation.

## 1. Introduction 

Single-cell RNA sequencing (scRNA-seq) uniquely characterizes tissue by dissecting the cellular heterogeneity at a high resolution. This methodology has proven to be indispensable in deepening our understanding of biological systems by uncovering intricacies of cell-to-cell variations, de novo cell populations, and differentiation trajectories [1]. Over the past decade, the application of scRNA-seq in both biological and clinical research has been immensely valuable, providing unexpected insights on the heterogeneity and functional specialization of tissues that were formerly thought to be homogenous. These investigative possibilities have broad implications for medical advancement through the identification of novel targets for therapeutic intervention. Single-cell approaches addressing adipose stromal cell heterogeneity have been concisely summarized by Rondini and Granneman [2]. Our review intends to cover developments in the singe-cell profiling of mature adipocytes, the application to human adipose tissues, and the most recent papers since the publication by Rondini and Granneman. Therefore, together, these reviews will provide the most up-to-date overview and comprehensive perspective on the impact of single-cell genomics on adipose tissue research.

Previously, cell-to-cell variation in adipose tissue was investigated using clonal analysis. For instance, Shinoda et al. demonstrated the functional heterogeneity of human brown adipose tissue (BAT) by identifying the distinct gene signatures of 65 clones [3]. Furthermore, Xue et al. defined novel gene expression signatures in human adipocyte progenitor cells (APCs) predictive of mature adipocyte thermogenic potential, which are functionally modified by genetic and spatial factors. Additionally, they identified marker genes, including CD29 (*ITGB1*), which aids in the prospective isolation of human APCs with a high thermogenic potential [4]. Later, Gao et al. identified four clones of APCs with distinct genetic signatures. They specifically categorized CD36 as a marker of APCs with a pronounced propensity to undergo differentiation and, when present in a high proportion, act as a marker of increased adipogenesis and reduced metabolic risk [5]. Using human adipose tissue explant [6], Min et al. identified three distinct mature human white adipose subtypes: (1) An adipocyte subtype specialized for lipogenesis, as denoted by the upregulation of *PPARγ*, *FABP4*, and *PLIN1*; (2) an adipocyte subtype sensitive to extracellular matrix (ECM) signaling, as indicated by the upregulation of genes associated with the ECM interaction and TGFβ signaling; and (3) a transcriptionally distinct, yet functionally unclassified, subtype. The distinction between these populations was further verified through differential adipokine production [7]. In mice, Lee et al. characterized three transcriptionally and metabolically discrete populations of white adipocytes, which were found to contribute variably to adipose depots, indicating functional specialization [8].

Although these findings laid a foundation for understanding adipocyte heterogeneity and functional specialization, it should be noted that single-cell cloning inevitably selects highly proliferative cell populations, regardless of the biopsied location and sample numbers. Therefore, direct single-cell transcriptome analysis is warranted to reveal the cellular diversity under more physiological conditions. Amongst the earliest of such studies was that by Korimoto et al., who pioneered a strategy for the amplification of mRNA from single cells for microarray analysis from mouse embryos [9], and subsequently cells undergoing germ cell specification [10]. ScRNA-seq without cell cloning was firstly documented in 2009 in a well-defined description of a single mouse blastomere [11]. Since then, arrays of new technologies have been developed that enable the profiling of single cells using next-generation sequencing [12,13,14,15,16]. As summarized in Figure 1, scRNA-seq technologies can be largely divided into two categories: (1) Plate- or microfluidics-based methods, in which the physical separation of cells into a compartmentalized space is required (e.g., Fluidigm C1), and (2) a droplet-based approach in which each a cell is trapped in hydrogel droplet-containing lysis buffer and assigned a DNA barcode (e.g., 10× Genomics Chromium). In both cases, the direct application to mature, lipid-containing adipocytes is challenging because adipocytes can easily rupture in microchips or during droplet formation. Accordingly, the initial application of scRNA-seq was to the non-adipocyte fraction of adipose tissues, including APCs and immune cells, which are summarized below.

## 2. Subsets of Adipocyte Progenitor Cells

Adipocyte progenitor cells (APCs) reside in the stromal vascular fraction (SVF) of adipose tissue, where they differentiate into mature adipocytes in response to external stimuli. APCs were typically defined by the lack of expression of hematopoietic/endothelial lineage markers (lineage-negative) and the high expression of mesenchymal stem cell markers, such as PDGFRα, CD34, and SCA-1 (*Ly6a*). The expansion of adipose tissue can occur through hypertrophy, an increase in adipocyte size, or hyperplasia, an increase in adipocyte number. Hyperplastic growth, achieved through the differentiation of APCs in a process known as adipogenesis, is central to the homeostatic adipose tissue function. When the local environment is subjected to deleterious signals secondary to metabolic imbalance, hypertrophic adipocyte growth results in inflammation and fibrosis, as well as the perpetuation of metabolic dysfunction [17,18]. As adipocytes are terminally differentiated, ensuing pharmacologic modulation will likely occur at the progenitor level, making the comprehensive characterization of APCs’ identity and functionality essential.

### 2.1. Adipogenesis-Regulatory (Areg) Cells

ScRNA-seq of mouse subcutaneous APCs was first reported by Schwalie et al., who applied this methodology to APCs defined as lineage-negative CD29*(Itgb1)*+CD34+SCA-1+ cells. They identified three distinct populations of APCs with a high expression of common progenitor or mesenchymal cell surface markers and an absence of mature adipocyte markers, substantiating their identity. Gene expression profiles demonstrated the distinct adipogenic functionality of each population, while the nuclei number remained consistent, verifying that adipogenic differences were not due to individual proliferative properties. The focus was then directed to the novel CD142(*F3*)+ABCG1+ APC population, which was found to negatively regulate the adipogenic capacity of other APCs, prompting its designation as ‘adipogenesis-regulatory’ cells, or ‘Aregs’. They discovered that this negative regulatory effect occurred through paracrine signaling dependent on *Rtp3*, *Spink2*, *Fgf12*, and *Vit*, and that the knockdown of these genes in Aregs or elimination Aregs entirely resulted in increased adipogenesis in surrounding populations. Notably, Aregs are functionally conserved in mouse visceral white adipose tissue (WAT) and human adipose tissue, with higher proportions in visceral fat and in all depots of obese individuals [19].

### 2.2. Fibro-Inflammatory Progenitors

Platelet-derived growth factor receptor beta (PDGFRβ or CD140B) is a marker for mural/smooth muscle-like APCs in adipose tissue. Hepler et al. isolated discrete populations of such PDGFRβ+ cells by performing scRNA-seq on mouse visceral WAT to explore the functional heterogeneity. Unsupervised clustering of the single-cell level transcriptome revealed three functionally distinct populations, which were independently validated using fluorescence-activated cell sorting (FACS): (1) LY6C- CD9- PDGFRβ+ cells, which are APCs with a high adipogenic potential, as well as committed preadipocytes; (2) LY6C- PDGFRβ+ cells, which are fibro-inflammatory progenitors (FIPs); and (3) visceral WAT-associated mesothelial cells. FIPs were a novel population, and thus further characterization was performed. They discovered that FIPs lack adipogenic capacity and even exert an anti-adipogenic effect on other APC subtypes [20]. While they share functional similarities with the CD142+ ‘Areg’ cells identified by Schwalie et al. [19], the cell types were determined to be molecularly distinct, as exemplified by FIPs’ unique fibrogenic gene expression profile and pro-inflammatory phenotype. Interestingly, the frequencies of these APC subtypes were found to be differentially regulated by high-fat diet (HFD) feeding, which resulted in a higher proportion of FIPs [20]. This data continues to highlight the functional heterogeneity of WAT APCs and provides insight on cell interactions impacting adipogenesis and fibrosis, which will facilitate the study of pathological WAT remodeling in vivo.

### 2.3. DPP4+ Multipotent Progenitors and ICAM1+ Committed Preadipocytes

To further investigate the progenitor cell hierarchy in adipose tissue, Merrick et al. performed scRNA-seq on mouse subcutaneous inguinal WAT and identified 10 distinct populations, including three major mesenchymal cell groups highly concordant with APCs identified in previous studies: (1) Highly proliferative DPP4+ multipotent mesenchymal progenitors; (2) CD54 (*Icam1)*+ committed preadipocytes; and (3) CD142+ adipogenesis-regulatory cells [17]. These APC cell types were molecularly conserved throughout all mouse adipose depots and in human adipose tissue. Following characterization, cellular trajectory analysis was performed, revealing that DPP4+ multipotent progenitors give rise to both CD54+ and CD142+ cells, which further differentiate into mature adipocytes. To maintain their progenitor identity and functional character, DPP4+ cells require TGFβ signaling, which mediates their proliferative and anti-adipogenic actions. Of note, obesity and insulin resistance were found to cause progenitor cell exhaustion and reduced precursor differentiation, specifically in visceral WAT, as depicted by DPP4+ APC depletion, which may contribute to pathological remodeling and metabolic disease progression. Therefore, targeting these cell populations may promote adaptive hyperplastic adipose growth to ameliorate metabolic disease [17].

### 2.4. Type 2 Diabetes Mellitus-Associated Adipocyte Progenitor Subtype

With the APC characterization of adipose tissue from normal-weighted mice and humans fairly well-established, Vijay et al. sought to identify disease-specific cell types in adipose tissue from obese individuals, which they obtained through collecting discarded samples from bariatric procedures. ScRNA-seq of WAT SVF revealed 17 total populations of cells: Seven populations of CD45*(Ptrpc)*-CD34+CD31*(Pecam1)*- APCs, seven populations of immune cells, and three populations of endothelial cells. Among the APCs, differential Adipsin (*CFD*) expression, a marker of adipocyte differentiation, demonstrated that some populations represented different stages of adipogenesis. The APC populations also had a striking depot-specific pattern and further analysis categorized cells into subcutaneous and visceral groups. The combined analysis revealed depot-independent cell types: (1) type II diabetes mellitus (T2D)-associated APCs; (2) fibroblasts; and (3) hematopoietic stem cells. Closer evaluation of the T2D-associated APCs revealed high levels of differentially expressed genes between adipose tissue from obese and normal-weight individuals, including *GPX3*, which was upregulated in cells from normal-weight individuals that negatively correlated with insulin resistance, while *WISP2 (CCN5)* and *AFT3* were upregulated in cells from T2D subjects and positively correlated with insulin resistance [21]. Overall, this data highlights the transcriptional variability of APCs based on the metabolic status in humans and contributes to the growing pool of considerations for the pharmacological modulation of adipose tissue to regulate metabolism. We summarized the aforementioned findings and papers on single-cell analysis during obesity and type 2 diabetes, as can be seen in Figure 2.

Although, in most circumstances, scRNA-seq has revealed a large heterogeneity within adipose tissue depots, one study by Acosta et al. on human APCs unexpectedly revealed the opposite. Acosta et al. analyzed APCs obtained from the cosmetic liposuction of subcutaneous WAT of healthy females using scRNA-seq and identified four distinct populations: a large cluster of APCs and three subtypes of macrophages. Applying pseudotemporal analysis to the APC population, they found slight fluctuations in marker genes regulated by adipogenesis; however, these differences were small and variable, which precluded separation into unique subpopulations. As such, they concluded that subcutaneous WAT APCs were a functionally homogeneous population [22]. However, the WAT APCs were obtained from different adipose tissue compartments than other studies (thigh and hips in cosmetic liposuction vs. omentum and abdomen in bariatric surgery), which should be taken into consideration during the interpretation of these findings.

## 3. Phenotypic Changes of Apcs during “Beiging”

The formation of brown/beige adipocytes in WAT is generally referred to as “beiging”, which involves the proliferation and differentiation of APCs in close interaction with resident immune cells. Burl et al. molecularly explored beiging through an investigation of thermogenic APCs using an in vivo model of beige adipogenesis to deconstruct adipogenic niches and map differentiation trajectories. To do so, they performed scRNA-seq of murine epididymal WAT (eWAT) and inguinal WAT (iWAT) during β3-AR (adrenergic receptor) activation by CL 316,243 (CL), which is a compound known to recruit beige adipocytes in WAT. Using PDGFRα and SCA-1 expression to isolate APCs, they discovered four populations in eWAT: (1) a large APC cluster with an upregulation of ECM-associated genes when activated; (2) a large APC cluster with an upregulation of cell motility-, migration-, and epithelial cell proliferation-associated genes when activated; (3) a small cluster of differentiating APCs enriched in genes involved in early adipogenesis; and (4) a small cluster of proliferating APCs with an inducible expression of genes that positively regulate the cell cycle. Amongst the differentiating and proliferating APCs, pseudotime analysis identified three major clusters that could be mapped to the known beiging trajectory. A corresponding analysis of iWAT identified two major APC populations, which were distinguished by a differential expression of genes involved in ECM production, proteolysis, and adipogenesis [23]. These findings characterize depot-selective APC heterogeneity and demonstrate that gene expression patterns are reflective of the cellular microenvironment.

Despite preliminary characterization, the developmental origin and regulatory pathways of beige adipocyte progenitors remained insufficiently understood. As such, Oguri et al. employed scRNA-seq of APC from mouse interscapular BAT (iBAT), iWAT, and eWAT [24]. They discovered at least five distinct cell populations, with one enriched in mural/smooth muscle-line gene sets previously implicated in beige adipogenesis, including *Sm22* and *Acta2,* and identified CD81 (*Tapa1*) as a highly-selective marker for the population [25,26]. The identity of these PDGFRα+SCA-1+CD81+ cells as beige APCs was confirmed using FACS and they subsequently demonstrated this population’s intrinsic plasticity and propensity to differentiate into beige adipocytes in vivo. Importantly, CD81+ was discovered to not only be a marker of beige APCs, but also critically functional in controlling beige fat biogenesis through the activation of focal adhesion kinase (FAK) signaling in response to irisin, which is a cytokine secreted by muscle that increases with exercise [27,28]. Using a mouse model depleted of this gene by CRISPR interference, Oguri et al. further showed that CD81 loss leads to metabolic dysregulation, including diet-induced weight gain, glucose intolerance, insulin resistance, and adipose tissue inflammation. Correspondingly, the CD81+APC population size in subcutaneous fat negatively correlated with metabolic health in humans. These findings indicate that CD81 is a useful marker for isolating beige APCs, as well as functioning as a key sensor of external inputs. We summarized the findings and papers on single-cell analysis during beiging, as can be seen in Figure 3.

### A Unified View of APC Heterogeneity

Due to pioneering applications of scRNA-seq of APC, it is now clear that there is tremendous heterogeneity amongst APCs beyond mesenchymal stem cell markers (PDGFRα, CD34, and SCA-1) and mural cell markers (PDGFRβ). In response to a variety of signals, such as obesogenic (high fat diet), pharmacological (CL-316,243), or cold adaptation (beiging), adipose tissue undergoes dramatic and dynamic remodeling. It is intriguing that some of the novel markers identified for APC subsets are required for their specialized function (e.g., Oguri et al.). Many APC markers (Sca-1, PDGFRα, CD29, or CD34) are expressed in stem cell populations of other tissues and are extensively studied outside the adipocyte field before their rediscovery in adipose tissue [29]. However, with the help of single-cell genomics and our community effort, it will now be possible to expand the use of novel markers identified from adipose tissues and “export” new biology to ever-growing stem cell research and regenerative medicine.

## 4. Cell Types of the Immune System Involved in Beiging and Obesity

Early transcriptomic analysis of bulk adipose-resident immune cells revealed their function in tissue remodeling and contribution to the pathogenesis of obesity. However, single-cell-level characterization of immune cell populations was required to define their roles with a higher resolution based on unbiased genome-wide gene expression.

### 4.1. Activated Macrophages during Β3-Agonist-Induced Beiging

Burl et al. performed the initial transcriptomic analysis of adipose-resident immune cells during their investigation of beige adipogenesis induced by the highly specific β3 adrenergic receptor (β3-AR) agonist CL 316,243 (CL). ScRNA-seq of mouse eWAT identified a population of natural killer T (NKT)-cells and two clusters of macrophages. With β3-AR activation, both macrophage populations expanded and transcriptionally segregated into subpopulations expressing a spectrum of M1- and M2-specific genes, which suggests that a range of macrophage activation states exist, defying traditional classifications. Within this spectrum, they discovered a set of macrophages with gene expression profiles enriched for proliferation, migration, adhesion, lipid uptake and metabolism, ECM remodeling and proteolysis, and foam cell differentiation. These functions are all essential for de novo beige adipogenesis, which begins with the clearance of dead white adipocytes and is followed by the recruitment of new APCs to the site of efferocytosis to guide their differentiation into beige adipocytes [23].

### 4.2. Lymphocyte Subtype Producing IL-10 upon Thermogenic Stimuli

Rajbhandari et al. discovered another functionally specific population of thermogenic adipose-resident immune cells in their evaluation of the role of immune cell-adipocyte crosstalk during thermogenesis. Using scRNA-seq of murine iWAT stromal vascular fractions, they identified 11 subpopulations of immune cells based on known marker genes: Four clusters of B cells, four clusters of T cells, and three clusters of macrophages. Having already proven that IL10 depletion promotes thermogenesis and confers diet-induced obesity resistance in mice [30], they sought to further dissect the role that immune cells play in this process. Upon thermogenic challenge, they witnessed an expansion in adaptive immune cell populations and upregulation of IL10 production in these subtypes, which antagonized thermogenesis, as hypothesized [31]. These studies illustrate the functional specialization of thermogenic adipose-resident immune cells and their dedicated role in thermal homeostasis.

### 4.3. Lipid-Associated Macrophage

While adipose-resident immune cells have been proven to contribute to metabolic disease pathogenesis, the molecular regulators mediating the immune cell remodeling of WAT during obesity remain largely unknown. As such, Jaitin et al. comprehensively mapped adipose-resident immune populations in obese mice and humans. First, they performed scRNA-seq of tissue-resident CD45+ (pan-hematopoietic marker) cells from mouse eWAT across a timeline of HFD feeding to capture lean and obese states and identified 15 populations, which displayed massive rearrangement in obese mice, specifically in macrophage subgroups. A distinct *CD9+Trem2*+ macrophage emerged exclusively under obese conditions, whose gene signature was enriched for lipid metabolism and phagocytosis. Hence, this population was named “lipid-associated macrophages” (LAMs). The LAM gene signature, specifically *Trem2*, was not found in other adipose tissue immune cells. Functional characterization discovered that Trem2 is not merely a characteristic marker, but also an essential driver of the LAM molecular program, as they showed that Trem2-knockout mice exhibit HFD-induced obesity, glucose intolerance, insulin resistance, and dyslipidemia. Importantly, the presence of TREM2-expressing LAMs was also confirmed in human visceral adipose tissue and the proportion of these cells correlated with the body mass index (BMI) [32].

### 4.4. Adipose-Resident T Cell Subtype Correlates with Human Obesity

The investigation of WAT-resident immune cells continued when Vijay et al. transcriptionally characterized the total cellular composition of the human WAT stromal vascular fraction. In addition to seven APC populations, they identified seven populations of immune cells, including T cells, B cells, NKT cells, dendritic cells, monocytes, and macrophages, which were further divided into 14 subpopulations based on unsupervised clustering. Next, they demonstrated the critical role that some of these cell types play in the loss of adipose homeostasis during obesity. One such cell type is a CCL5+CD8+ T cell that differentially expresses metallothionein genes, which have a strong association with the obesity phenotype. They established that these obesity-activated T cells recruit CD9+ adipose-resident macrophages, which are highly metabolically active and involved in lipid accumulation and trafficking [21]. These findings are consistent with the identity and functionality of Jaitin et al.’s LAMs and corroborate their conclusions that obesity activates an inflammatory macrophage phenotype [21,32]. Together, these studies began to uncover the central role that adipose-resident immune cells play in the pathogenesis of metabolic disease, and suggested potential means for the pharmacologic exploitation of key molecular mediators in the development of efficacious interventions.

## 5. Challenges and Promises in Dissecting Mature Adipocyte Heterogeneity

As aforementioned, the direct application of scRNA-seq to mature, lipid-containing adipocytes has been challenging due to the large size and high buoyancy of white adipocytes. To overcome this, researchers have taken three different approaches: (1) Harvesting single mature brown adipocytes by precise pipetting, followed by the generation of an RNA-seq library [33]; (2) limiting their analysis to relatively small thermogenic adipocytes and then running scRNA-seq [34]; and (3) isolating nuclei and running single nuclei RNA-sequencing [31]. We will review each of these three approaches below.

### 5.1. Physical Isolation of Mature Brown Adipocytes

Brown adipocytes (BAs) are mitochondria-rich cells specialized for adaptive non-shivering thermogenesis, and play a crucial role in protecting against obesity and metabolic disease. However, other than β3-AR stimulation, information about alternate activating signals remains unknown. As such, Spaethling et al. used RNA-seq of nine single mature BAs isolated from mouse brown adipose tissue (BAT) for the cellular characterization and identification of functional targets directly activating thermogenesis. This investigation demonstrated immense cellular heterogeneity amongst BAs, including variability in the expression of marker genes, such as *Ucp1*, *Adra1a*, and *Adrb3*. In addition to known marker genes, numerous orphan fatty acid transporters, including the *Slc27a* family, as well as various glutamate, amino acid, and ion transporters, were discovered. This newly defined repertoire increased the catalogue of recognized BA transporters and channels from 20 to > 500. Moreover, they identified 266 lncRNAs, non-protein-encoding RNAs that exert regulatory control over other RNA molecules, which were variably expressed among BAs. Given the variability of marker, receptor, and lncRNA expression, it can be assumed that BAs function heterogeneously and receive and emit unique signals as dictated and required by their transporters and channels [33]. 

### 5.2. Low- and High-Thermogenic Brown Adipocytes

While it was determined that functional heterogeneity exists amongst BAs, further investigation on a larger scale (hundreds to thousands of mature adipocytes) was required to better understand BA functional heterogeneity at the cellular level. Song et al. accomplished this task through applying scRNA-seq of BAT [34]. In this study, four distinct cell populations were uncovered: Two populations of BAs, one population of white adipocytes (WA), and a non-adipocyte cluster. Transcriptional profiling of the BA subpopulations revealed that one was the classical, well-studied, highly thermogenic BA; however, the other displayed substantially lower thermogenic activity, as denoted by the low expression of *Ucp1* and *Adipoq* and low mitochondrial content. These low-thermogenic BAs were functionally assessed and determined to be enriched in genes for fatty acid uptake, cell-to-cell trafficking, and *Ucp1*-independent thermogenesis. With cold-temperature exposure, these BA subpopulations underwent dynamic interconversions between low-thermogenic BAs and high-thermogenic BAs, which reversed when a thermoneutral environment was restored. This cellular interconversion was not affected by HFD-feeding, but did decline with age [34]. These findings further characterize the molecular heterogeneity of BA subpopulations and support new strategies, such as promoting the conversion of low-thermogenic BAs into high-thermogenic BAs, for the possible pharmacological management of metabolic disease.

### 5.3. Single-Nuclei RNA-Seq

Recent literature comparing well-matched single-nuclei RNA-sequencing (snRNA-seq) and scRNA-seq datasets has revealed that snRNA-seq is comparable to scRNA-seq in cell-type detection and superior in the cellular coverage for complex tissues, such as the brain, despite the known loss of mRNA from the cytoplasm and other organelles [35]. With this confidence in its efficacy, Rajbhandari et al. characterized beige adipocytes via snRNA-seq of mouse inguinal WAT. They identified 14 cell populations with unique transcriptional profiles descriptive of specialized biological functions. One population was enriched with genes associated with fatty acid metabolism, which is indicative of a highly metabolically active population. The concomitant presence of known thermogenic marker genes confirmed this population’s identity as beige adipocytes. Based on their previous findings that mice lacking IL10 exhibit enhanced thermogenesis and are protected from diet-induced obesity, they looked for this beige adipocyte population in IL10 receptor knockout mice and expectedly found it in a high proportion [31]. Overall, the transcriptional dissection of mature beige adipocytes illustrates the important role of these cells in maintaining a healthy metabolic index and sets the foundation for further research on the pharmacologic inducibility of beige adipocytes by modulating cytokine signaling.

### 5.4. Caveats in Profiling Mature Adipocyte Heterogeneity

As discussed, the profiling of mature adipocyte heterogeneity remains challenging and methodologic optimization is ongoing in several laboratories. One major issue with current protocols is the potential disruption of nuclei during isolation procedures, which allows for mRNA leakage, increasing the background signal and decreasing the number of recoverable nuclei, as well as reducing the depth of reads. To combat these issues, Benitez et al. have developed an additional step in the snRNA-seq protocol, which uses a gentle MoFlo XDP sorter to limit cytosolic RNA contamination and nuclear aggregation to produce a cleaner sample that better characterizes each cell individually, as well as adipocyte heterogeneity among the entire sample [36]. In addition to this important methodological issue, there are also caveats with data processing. As a majority of RNA molecules in the nucleus are unprocessed “nascent” RNA, using an exon-only reference library for read alignment, which is the default setting in most single-cell gene expression software, will result in losing information on hundreds of otherwise-detectable genes. This can be avoided by creating a “pre-mRNA” reference package, which includes intronic regions.

## 6. Limitations and Future Directions

Pioneering single-cell technologies are uncovering novel and critical information on tissue heterogeneity and function; however, it is important to recognize the current shortcomings of these approaches. Differences in experimental models and adipose tissue depots, as well as the bioinformatics tools applied to these data sets, must all be taken into account when assessing the validity of findings and their interpretations. One limitation specific to all scRNA-seq is the exclusive ability to measure mRNA with the loss of important information from proteins and their post-translational modifications, which requires other methodologies, such as flow cytometry using labeled antibodies. Measuring two or more modalities simultaneously (e.g., protein and transcriptome), made possible by single-cell multimodal omics techniques [37,38], would enable a more complete understanding of the cellular and molecular heterogeneity in adipose tissue and allow the identification of novel subtypes. It is also worth mentioning the importance of combining scRNA-seq with the Assay for Transposase-Accessible Chromatin (scATAC-seq) in studying the adipocyte epigenome and transcriptome simultaneously given the importance of epigenetic reprogramming in downstream gene expression and adipocyte function. Such technology was recently made possible by physically separating genomic DNA and mRNA and was validated using cancer cells [39].

Interpretational capabilities could be further enhanced if the in vivo function of unique/rare cell subtypes could be tested by isolation and transplantation. Sophisticated single-clone transplantation assays exist in other fields, such as hematopoietic research, which permit clonal single-cell transplantations for the in vivo investigation of cell type functionality. In contrast, this is not yet possible with current methods for adipose tissue research that requires the transplantation of millions of cells with low effective engraftment rates.

The remarkable capabilities of single-cell genomics provide distinctive opportunities for future research to address these limitations and expand the applications of these technologies. For instance, established methodologies require the disassembly of specimens at the cost of disrupting the tissue morphology; thus, improved technologies will permit analogous transcriptomic analysis from intact histological tissue specimens, which could allow for the localization of findings within the studied tissue. Additionally, the broad application of multiplex technology, either through natural genetic variation [40], chemical labeling [41,42], or a DNA-tagged antibody [43,44], will further improve scRNA-seq and snRNA-seq capabilities by increasing the throughput to minimize batch effects, reduce costs, and streamline the preparation of large numbers of samples.

In the authors’ opinion, an essential application of scRNA-seq to adipose tissue research will be the ability to methodically phenotype tissue-specific knockout mice. For instance, Adiponectin-Cre recombinase [45,46] and UCP1-Cre recombinase [47] have been widely used to study a variety of genes, including receptors, enzymes, and transcription factors. Applying the abovementioned technology to adipose-specific knockout mice will provide information on subtype-specific transcriptome changes within Cre-driver+ cells, as well as the paracrine effect on neighboring Cre-negative cells, thus facilitating a systematic understanding of the function of genes and interplay of cell types, which is difficult to recapitulate in vitro. Reanalyzing previously reported adipose-specific knockout mice using scRNA-seq and snRNA-seq could lead to the discovery of novel cell types or paracrine effects that better explain associated metabolic phenotypes (e.g., diet-induced obesity, glucose intolerance, insulin sensitivity, and hypothermia).

## 7. Conclusions

The unique capabilities of single-cell genomics have contributed a wealth of knowledge to adipocyte research and the understanding of adipose tissue’s role in the regulation of whole-body metabolism. The transcriptomic dissection of mouse and human adipose tissue has comprehensively classified white and thermogenic adipocytes, various APC subtypes, and adipose-resident immune cells. These findings have elucidated the composition and functional specialization of numerous distinct populations, their marker genes, and their role in tissue homeostasis or loss thereof. A multitude of molecular regulators centrally implicated in metabolic health have been identified, and further research is needed to determine whether these targets are suitable for safe and efficacious pharmacologic manipulation to selectively manage intrinsic metabolic dysfunction.

## Figures and Tables

**Figure 1 ijms-21-04773-f001:**
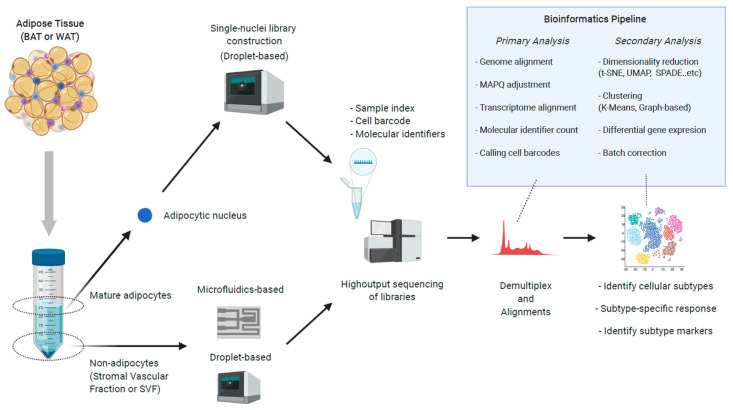
The workflow of single-cell-level RNA-sequencing of Brown Adipose Tissue (BAT) and White Adipose Tissue (WAT).

**Figure 2 ijms-21-04773-f002:**
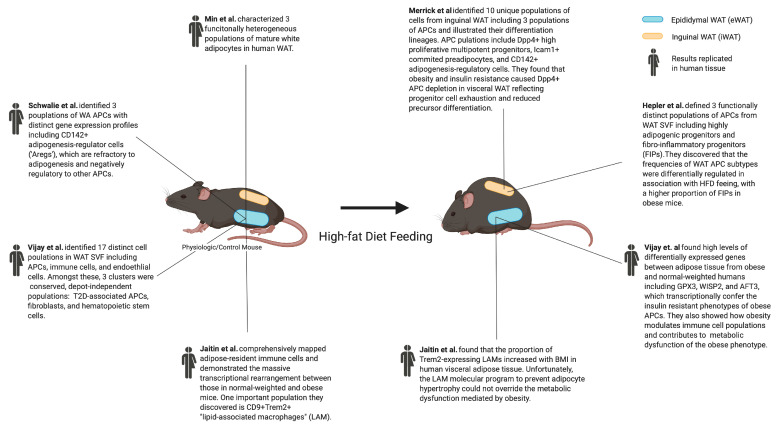
Single-cell genomics uncovers cellular heterogeneity in the adipose tissue and phenotypic changes during obesity.

**Figure 3 ijms-21-04773-f003:**
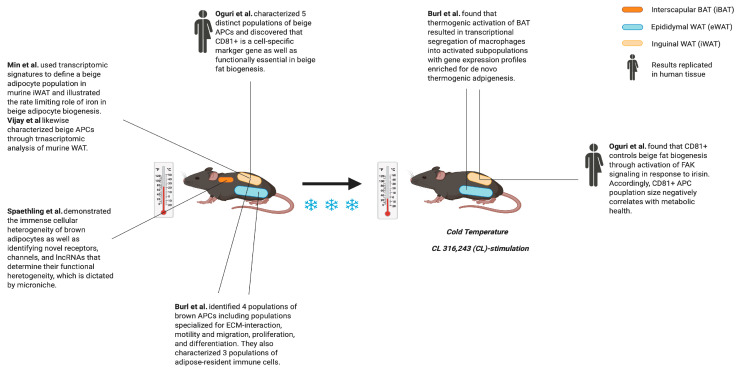
Single-cell genomics identifies a subset of adipocyte precursor cells (APCs) and molecular regulators critical for the beiging of White Adipose Tissue (WAT).

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
