# Peer review of "The Impact of Single-Cell Genomics on Adipose Tissue Research"

_ijms, 2020, doi:10.3390/ijms21134773_

Round 1

Reviewer 1 Report

Title: The Impact of Single Cell Genomics on Adipocyte Research

Authors: Alana Deutsch, Daorong Feng, Jeffrey E Pessin and Kosaku Shinoda.

General Opinion

Single cell analysis is an important trend in the study of adipose tissue physiology that helps reveal the complexity of the cells that make up this tissue. That is why I consider a review made by Alan Deutsch et al. very interesting because it tries to sum up the research to date in this field. I have only a few minor remarks that do not undermine the substantive value of work.

Minor revisions

  1. The authors should pay attention to the fact that every attempt to perform scRNA-seq in cells isolated from the adipose tissue leads to the discovery/description of new cell phenotypes. This finding may refer to a real phenomenon, but it is also possible that methodological issues are responsible for this multiplicity. For instance, the differences in experimental models (murine vs. human), adipose tissue depot (epididymal/inguinal in mice, subcutaneous gluteofemoral /subcutaneous abdominal/visceral in humans) as well as bioinformatics tools applied to analyse the data from the wet lab.

  1. I do appreciate excellent figures that summarize the studies described in each chapter of the review. However, the part entitled Subsets of Adipocyte Progenitor Cells misses a summary, highlighting the data that may constitute a “common denominator” of the presented research.

  1. Since the review also covers studies regarding the immune cells infiltrating adipose tissue, it would be worth to consider modification of the title from The Impact of Single Cell Genomics on Adipocyte Research to The Impact of Single Cell Genomics on Adipose Tissue Research.

  1. Lines 270-271: “Harvesting single mature adipocytes by precise pipetting followed by generation of an RNA-seq library”- since the following paragraph refers to the mature brown adipocytes, adding “brown” (“single mature brown adipocyte”) would clarify that this chapter refers to beige/brown adipocytes.
  2. Line 273 “….(I just switch the order to match following content)….” – the sentence in parentheses seems to come from the draft of the manuscript.

Author Response

  1. The authors should pay attention to the fact that every attempt to perform scRNA-seq in cells isolated from the adipose tissue leads to the discovery/description of new cell phenotypes. This finding may refer to a real phenomenon, but it is also possible that methodological issues are responsible for this multiplicity. For instance, the differences in experimental models, adipose tissue depot, as well as bioinformatics tools applied to analyze the data from the wet lab.

We greatly appreciate this comment, and others below, which have influenced the authors in adding a limitations and future steps section to our manuscript. We discuss the issues addressed in this comment in Section 6, paragraph 1, lines 357-361.

  1. I do appreciate excellent figures that summarize the studies described in each chapter of the review. However, the part entitled ‘Subsets of Adipocyte Progenitor Cells’ misses a summary, highlighting the data that may constitute a “common denominator” of the presented research.

In response to this comment, as well as Reviewer 4’s comment 2, we have added a summary entitled “A Unified View of APC Heterogeneity”, see lines 211. Thank you for this suggestion.

  1. Since the review also covers studies regarding the immune cells infiltrating adipose tissue, it would be worth to consider modification of the title from ‘The Impact of Single Cell Genomics on Adipocyte Research’ to ‘The Impact of Single Cell Genomics on Adipose Tissue Research’.

We have changed the title, as recommended by Reviewer 1. We agree it better represents the content of the entire review. Thank you!

  1. Lines 270-271: “Harvesting single mature adipocytes by precise pipetting followed by generation of an RNA-seq library”- since the following paragraph refers to the mature brown adipocytes, adding “brown” (“single mature brown adipocyte”) would clarify that this chapter refers to beige/brown adipocytes.

We agree adding ‘brown’ more appropriately introduces/reinforces the topic of the chapter and have made the recommended change, accordingly. See lines 287.

  1. Line 273 “…(I just switch the order to match following content)…” – the sentence in parentheses seems to come from the draft of the manuscript.

We sincerely apologize for leaving this note in our submitted manuscript. It has been removed.

Reviewer 2 Report

Summary

In this review article, Deutch et al. summarize the recent technical advancement in single cell gene expression analyses, and overviews literatures that utilized these methods to understand the cellular heterogeneity of adipose tissues. This review covers almost all the papers that used scRNA-seq in adipose tissues to date, and for sure will be an informative source for readers in the metabolism field, especially for those studying adipose tissue cellular heterogeneity and also for those looking for adipose tissue scRNA-seq datasets. The review is nicely written, and will be beneficial to people studying the adipose tissues for reasons mentioned above. On the other hand, it also feels like this review is a list of published paper abstracts and somehow do not contain a unified view of the cellular heterogeneity that has been uncovered using the scRNAseq techniques. I would be more pleased if the authors could provide some points listed below, such as summary of current knowledge in biology of APC heterogeneity, current challenges for further understanding of the tissue functions, and future directions in the field.

Comments:

  1. Historically, flow-cytometer based cell surface marker profiling has been widely used in the hematopoietic field to identify distinct population of cells through the use of available antibodies. Hundreds of antibodies have been tested to see whether they show a heterogenous staining pattern. In a sense, scRNA-seq does the same thing with mRNAs in a transcriptome-wide fashion. This advancement is remarkable, however, it would also be interesting to know the caveats and difficulties of the techniques. For example, cell isolation methods used that may create biases, coverage of cell populations and further possibilities of identifying a new types/ states within adipose tissues.

  1. Given that the scRNA-seq profiling of adipose tissues have been utilized in various literatures mentioned above, it would be nice to hear the authors summary (or unified view) of the understanding of APC heterogeneity and functions in one section.

  1. As mentioned in section 5, profiling of mature adipocyte heterogeneity still sounds challenging and unsolved. What are the authors opinion in approaches in characterization of mature adipocyte heterogeneity (both technically and biologically)? What were the caveats for each literatures? Are they robust enough to conclude what the paper claims? What are the issues needed to be addressed?

  1. It would be interesting to know the author’s opinion/ interpretation on further utilities of scRNA-seq once the characterization of the normal adipose tissue heterogeneity is achieved. In other words, what are the unsolved biological issues and questions that can be addressed using the single cell genomics? Applying scRNA-seq to perturbed conditions (either metabolically or genetically) seems like a straightforward approach, but are there any other ways to advance our understanding using this technique?

Minor comments:

  1. It has always been difficult to assess the biological relevance of a unique (and especially rare) cell population identified. As far as I understand, these obstacles still exist in the adipose tissue field. In a lot of cases this can only be assessed in vivo. (For example, the hematopoietic stem cell field heavily relied on transplantation assays to assess the function of a newly identified cell population). It would be important to mention that establishment of these assays are necessary to understand the function of a new cell type identified through single cell genomic profiling.

  1. Single cell genomics were present before 2009 paper (Reference#8) the authors cited, for example the use of microarray after amplification of cDNAs from single cells isolated from early mouse embryos (2006 Kurimoto et al.) and subsequently in cells undergoing germ cell specification (2008 Kurimoto et al.). For precision, the authors can either cite these papers or refer to the use of “single cell RNA-seq” instead of “single cell genomics”.

Author Response

  1. Historically, flow-cytometer based cell surface marker profiling has been widely used in the hematopoietic field to identify distinct population of cells through the use of available antibodies. Hundreds of antibodies have been tested to see whether they show a heterogenous staining pattern. In a sense, scRNA-seq does the same thing with mRNAs in a transcriptome-wide fashion. This advancement is remarkable, however, it would also be interesting to know the caveats and difficulties of the techniques. For example, cell isolation methods used that may create biases, coverage of cell populations and further possibilities of identifying a new types/ states within adipose tissues.

Thank you for this recommendation that was also suggested by the other reviewers. We have added Section 6. Limitations and Future Directions, which cover the authors’ views on limitations in current single cell-based methodological approaches and data interpretation as well as the future directions of single cell genomics in adipocyte biology research. See lines 356.

  1. Given that the scRNA-seq profiling of adipose tissues have been utilized in various literatures mentioned above, it would be nice to hear the authors summary (or unified view) of the understanding of APC heterogeneity and functions in one section.

In response to this comment, as well as Reviewer 1’s comment 2, we have added a summary for Section 3.1 A Unified View of APC Heterogeneity, see lines 211. Thank you for this suggestion and we agree it provides the discussion necessary to present the authors’ unified view on the understanding of APC heterogeneity.

  1. As mentioned in section 5, profiling of mature adipocyte heterogeneity still sounds challenging and unsolved. What are the authors opinion in approaches in characterization of mature adipocyte heterogeneity (both technically and biologically)? What were the caveats fro each literature? Are they robust enough to conclude what the paper claims? What are the issues needed to be addressed.

We agree that discussion of the challenges in approaches in characterization of mature adipocyte heterogeneity is important. As such, we have added Section 5.4 Caveats in profiling mature adipocyte heterogeneity, which shares the authors’ viewpoint on the subject.

  1. It would be interesting to know the author’s opinion/ interpretation on further utilities of scRNA-seq once the characterization of the normal adipose tissue heterogeneity is achieved. In other words, what are the unsolved biological issues and questions that can be addressed using the single cell genomics? Applying scRNA-seq to perturbed conditions (either metabolically or genetically) seems like a straightforward approach, but are there any other ways to advance our understanding using this technique?

Thank you for this comment and we have added the authors’ views on limitations in current single cell-based methodological approaches and the future directions of single cell genomics in adipocyte biology research. See lines 387-397.

  1. It has always been difficult to assess the biological relevance of a unique (and especially rare) cell population identified. As far as I understand, these obstacles still exist in the adipose tissue field. In a lot of cases this can only be assessed in vivo. (For example, the hematopoietic stem cell field heavily relied on transplantation assays to assess the function of a newly identified cell population). It would be important to mention that establishment of these assays are necessary to understand the function of a new cell type identified through single cell genomic profiling.

This concept of assessed the biological relevance of a unique cell population identified by scRNA-sec is extremely important and still stands as an obstacle in adipose tissue research. We have included discussion of this issue in newly added Section 6. Limitations and future directions. See lines 372-377.

  1. Single cell genomics were present before 2009 paper (Reference#8) the authors cited, for example the use of microarray after amplification of cDNAs from single cells isolated from early mouse embryos (2006 Kurimoto et al.) and subsequently in cells undergoing germ cell specification (2008 Kurimoto et al.). For precision, the authors can either cite these papers or refer to the use of “single cell RNA-seq” instead of “single cell genomics”.

Thank you for making us aware of this inaccuracy that we have now addressed in the introduction section. See lines 60-62.

We wish to again thank the editor and reviewers for your time and effort in consideration of our manuscript. We are enthusiastic about the opportunity for publication in IJMS.

Reviewer 3 Report

The Impact of Single Cell Genomics on Adipocyte Research.

Deutsch et al., describes the use of scRNA-seq and single-nuclei RNA-seq (snRNA-seq) in adipocyte biology and its applications to obesity and diabetes research. Recently, great progresses have been achieved in single cell sequencing, which stands out among all single cell analysis techniques. The progression of next generation sequencing is continuously changing the landscape of genomic, transcriptomic, and epigenomic studies. Particularly, advances in single cell manipulation and amplification techniques bring sequencing technology to the single-cell level. I appreciate the authors for chosen this topic and described the use of scRNA-seq and snRNA-seq in adipocyte biology research. However, this study has limitations and lack of information regard to diabetes research.

The relevance of the applications is clearly addressed in adipocyte biology. I hope authors may include the study of limitation and future directions of scRNA-seq and snRNA-seq in adipocyte biology research.

Overall concept is good and manuscript is good in shape. Before going to publication may check some typo graphical errors and misspellings. I would say this manuscript is acceptable and publishable.

Author Response

The relevance of the applications is clearly addressed in adipocyte biology. I hope authors may include the study of limitation and future directions of scRNA-seq and snRNA-seq in adipocyte biology research.

Thank you for this suggestion, which was also indicated by the other reviewers. We have added new Section 6. Limitations and Future Directions, which cover the authors’ views on the future directions of single cell genomics in adipocyte biology research. See lines 356.

Reviewer 4 Report

20200618_IJMS_Deutsch et al

The present manuscript by Deutsch et al provides a good overview of findings in adipocyte research using single cell genomics. While some of these topics are discussed in a recent review (Rondini et al., in Biochemical Journal feb 2020), the present manuscript describes various cell types identified using single cell approaches, the markers genes defining adipose tissue cell types,  their spatial organization, and their functional roles in physiological and  pathogenic states, such as obesity and diabetes, and in thermogenesis.

I have a few minor suggestions:

1. Figures 2 and 3 essentially are sentences already mentioned in the text. The font size is too small to read. It would be very helpful to the reader if they use a visual representation (e.g., illustrate spatial organization of tissue with cell types colored according their new classification or function and list their respective markers) or a table (e.g., author name, tissue examined-epdidymal/inguinal, number of cell types identified, markers genes, human/mouse, healthy/obese or ambient temp/thermogenesis, etc) to summarize these various findings in literature.

2. Most of the genes listed have alternative names. It might be helpful to include the latest gene names in parentheses (e.g., Sca1 (Lys6a) and for most CD molecules, CD29 (Itgb1)). 

3. While much of the review focusses on transcriptomics, it might worthwhile to discuss the importance of epigenetics (methylome) given their role in reprogramming of adipocyte function in obesity in future directions.

4. Also, the authors should a note how the present manuscript is different from the recent one by Rondini et al.

5. On line 273, delete “(I just switch the order to match following content)”.

Author Response

  1. Figures 2 and 3 essentially are sentences already mentioned in the text. The font size is too small to read. It would be very helpful to the reader if they use a visual representation (e.g., illustrate spatial organization of tissue with cel types colored according their new classification of function and list their respective markers) or a table (e.g., author name, tissue examined- epididymal/inguinal, number of cell types identified, marker genes, human/mouse, healthy/obese or ambient temp/thermogenesis, etc) to summarize these various findings in literature.

We greatly appreciate this suggestion. We have increased the font size for the readers. Although table representation is useful, it was done by Rondini and Granneman in their review (Table 1 in reference 2). As such, we prefer to keep the original style of the figures so that both representations can provide perspectives on adipose single cell genomics for broad readerships. Thank you for this suggestion.

  1. Most of the genes listed have alternative names. It might be helpful to include the latest gene names in parentheses.

For all genes with alternative names, we have included that name in parentheses next to alternate name the first time it appears in the manuscript.

  1. While much of the review focuses on transcriptomics, it might worthwhile to discuss the importance of epigenetics (methylome) given their role in reprogramming of adipocyte function in obesity in future directions.

Thank you for this suggestion. We have added comments on the importance of epigenetics in the newly-added Section 6. Limitations and Future Directions. See lines 366-371.

  1. Also, the authors should note how the present manuscript is different from the recent one by Rondini et al.

Thank you for making us aware of this inaccuracy. We have added comment and citation of the manuscript in our introduction. See lines 32-37.  

  1. On line 273, delete “(I just switch the order to match following content)”.

We sincerely apologize for leaving this note in our submitted manuscript. It has been removed.

We wish to again thank the editor and reviewers for your time and effort in consideration of our manuscript. We are enthusiastic about the opportunity for publication in IJMS.